# A Real-Time Fluorescent Reverse Transcription Quantitative PCR Assay for Rapid Detection of Genetic Markers’ Expression Associated with Fusarium Wilt of Banana Biocontrol Activities in *Bacillus*

**DOI:** 10.3390/jof7050353

**Published:** 2021-04-30

**Authors:** Shu Li, Ping He, Huacai Fan, Lina Liu, Kesuo Yin, Baoming Yang, Yongping Li, Su-Mei Huang, Xundong Li, Si-Jun Zheng

**Affiliations:** 1Yunnan Key Laboratory of Green Prevention and Control of Agricultural Transboundary Pests, Agricultural Environment and Resources Institute, Yunnan Academy of Agricultural Sciences, Kunming 650205, China; lishukm@yeah.net (S.L.); heping_superv@163.com (P.H.); hcfan325@126.com (H.F.); handyliu@126.com (L.L.); yinkesuo@163.com (K.Y.); yangbaoming1964@126.com (B.Y.); km_lyp1443@126.com (Y.L.); xundonglee@sina.com (X.L.); 2State Key Laboratory for Conservation and Utilization of Bio-Resources in Yunnan, Ministry of Education Key Laboratory of Agriculture Biodiversity for Plant Disease Management, College of Plant Protection, Yunnan Agricultural University, Kunming 650201, China; 3Biotechnology Research Institute, Guangxi Academy of Agricultural Sciences, Nanning 530007, China; hsmei74@126.com; 4Bioversity International, Kunming 650205, China

**Keywords:** *Bacillus*, biocontrol, *Foc* TR4, biocontrol marker genes, PCR, RT-qPCR, biocontrol mechanisms, *Bacillus* RNA extraction

## Abstract

Fusarium wilt of banana, caused by *Fusarium oxysporum* f. sp. *cubense* (*Foc*), especially Tropical Race 4 (TR4), seriously threatens banana production worldwide. There is no single effective control measure, although certain *Bacillus* strains secrete antibiotics as promising disease-biocontrol agents. This study identified five *Bacillus* strains displaying strong antibiotic activity against TR4, using a systemic assessment for presence/absence of genetic markers at genome level, and expression profiles at transcriptome level. A conventional PCR with 13 specific primer pairs detected biocontrol-related genes. An accurate, quantitative real-time PCR protocol with novel designed specific primers was developed to characterise strain-specific gene expression, that optimises strain-culturing and RNA-isolation methodologies. Six genes responsible for synthesising non-ribosomal peptide synthetase biocontrol metabolites were detected in all five strains. Three genes were involved in synthesising three Polyketide synthetase metabolites in all five strains, but the macrolactin synthase gene *mln* was only detected in WBN06 and YN1282-2. All five *Bacillus* strains have the genes *dhb* and *bioA*, essential for synthesising bacillibactin and biotin. However, the gene *sboA,* involved in subtilisin synthesis, is absent in all five strains. These genes’ expression patterns were significantly different among these strains, suggesting different mechanisms involved in TR4 biocontrol. Results will help elucidate functional genes’ biocontrol mechanisms.

## 1. Introduction

Banana (*Musa* spp.) is a globally important tropical and subtropical crop. Globally, as the second largest fruit crop with the largest world fruit trade, and the fourth largest food crop after rice, wheat and maize, more than 400 million people depend on it as their staple food [1,2]. Fusarium wilt of banana (FWB)—also known as Panama disease—caused by *Fusarium oxysporum* f. sp. *cubense (Foc)*, especially Tropical Race 4 (TR4), seriously threatens banana industries in China and across the globe. Due to its virulence factor, TR4 isolates are designated as vegetative compatibility group VCG 01213 (or VCG 01216), which originates from Indonesia and affects many banana varieties [3,4]. This disease is known as “banana cancer” because of its destructive and highly pathogenic characteristics. Commercially-grown bananas are mainly from the Cavendish subgroup, with a uniform genetic background and lacking TR4-resistant cultivars. Rotation is a partially effective complementary FWB control measure, but not practical in large-scale monocultures, where extensive chemical pesticide use degrades natural environments. Therefore, development and application of bacterial reagent-based biological control agents for banana diseases are greatly anticipated [4,5,6].

Amongst biocontrol agents against plant disease, the bacterial *Bacillus* group displays several advantages over non-*Bacillus* species, because of its spore-forming ability. Spores can survive in some harsh environments. Moreover, spores have more stability in commercial production and resistance against chemical pesticides, etc. [4,7]. There are many kinds of biocontrol agents in the *Bacillus* group that could be used against FWB. So far, there is only one *Bacillus subtilis* bioagent (Registration No. PD20101654,) registered in China’s pesticide-registration database of its pesticide information network (http://www.chinapesticide.org.cn/hysj/index.jhtml, accessed on 6 February 2021). In addition to the registered *B. subtilis*, other *Bacillus* strains with some proven biocontrol effect against TR4 have been reported, such as (i) *B. subtilis* strains N11 [8], B26 [9], B04, B05 and B10 [10], (ii) some *B. velezensis* strains, including HN03 [11] and (iii) *B. amyloliquefaciens* strains, including: NIN-6 [12], NJN-6 [13] and W19 [14].

In recent years, several TR4 antagonistic *Bacillus* strains have been screened and identified, which provides a new basis for FWB biological control.

There are mainly two types of the biocontrol mechanisms for *Bacillus*: direct antagonism and indirect antagonism. Direct antagonism includes competition (nutrition competition and spatial site competition) and synthesis of secondary metabolites with antibiosis activity, hydrolases and release of volatile organic compounds. Indirect antagonism includes inducing plant resistance and inhibiting pathogens by affecting the diversity of soil or plant microbial community [4].

The antagonistic substances produced by *Bacillus* can be divided into two main categories. One is the cyclic lipopeptides synthesized by non-ribosomal peptide synthetase (NRPS), among which surfactin [15,16], iturins [17,18] and fengycin [19,20] are the most important antibioses, where *srfAA*, *ituC* and *fenD* are the key synthetase genes of these three antibioses, respectively. For lipopeptides subtilisin A and bacillibactin, the key synthetase genes are *sboA* and *dhbA*, respectively. In addition, two yndJ and yngG proteins are included in the lipopeptide compounds, and their corresponding synthetase genes are *yndJ* and *yngG*, respectively. In addition, these genes were used as marker genes for biocontrol [4,21,22]. The conventional PCR method for characterizing functional genes can be used to quickly detect and evaluate the potential of their corresponding antibiotics’ syntheses [21], while the real-time fluorescent quantitative RT-PCR method can be used to obtain the expression of functional genes [22]. That will further provide useful clues for elucidating biological control mechanisms. However, current marker genes are not sufficient to categorize all biocontrol agents.

Therefore, in this study, we would like to apply microscopic observation on the TR4 mycelial morphology during dual-culturing of five strains: WBN06, HN04, YN1282-2, N67 and G9R-3, which were isolated from two mainly banana-producing provinces of Yunnan and Guangxi. The results could provide the initial pointers for research on *Bacillus* biocontrol mechanisms. Secondly, a conventional PCR method with thirteen pairs of specific biocontrol marker genes including seven NRPS genes, four PKS genes and two RPS genes in the genus *Bacillus* will be used to detect five strains of *Bacillus*, and their biocontrol potential of the above-mentioned five strains was evaluated in vitro. We further plan to develop an improved method suitable for extracting high-quality *Bacillus* RNA which could provide a reliable technical support for RT-qPCR quantification. At the end, we would like to develop a real-time fluorescence quantitative detection system to detect 12 biocontrol marker-genes in *Bacillus* that confer TR4 antagonism. We hope that the results could provide an important theoretical basis for further study of gene functions and their biological control mechanisms.

## 2. Materials and Methods

### 2.1. Fungal Strain and Bacillus Strains

The highly pathogenic *Foc*TR4 strain (15–1) was used in this study [23]. The *Bacillus* strains N67, WBN06, HN04 and G9R-3 were isolated from Guangxi banana plantations and YN1282-2 was isolated from Yunnan banana tissue-cultured plantlets. The fungal and *Bacillus* strains were stored in liquid medium with 15% glycerol at −80 °C.

### 2.2. Dual-Culture Test

The scale lines on a PDA medium with a base plate diameter of 90 cm were marked. The scale lines include two straight lines that run at right angles through the centre of the upper surface and are positively intersecting. A hole-puncher was used to extract an agar disk containing TR4 mycelia with a diameter of 0.7 cm, taken from the edge of the TR4 colony, and placed at the centre of the scale intersection. The *Bacillus* strain monoclonal colony was selected with a toothpick and inoculated on a scale line 2.5 cm away from the TR4 mycelium disk. The process is shown schematically in Figure 1a. TR4 was inoculated at the centre of the scale intersection, and the *Bacillus* strain inoculated on a scale line at a distance of 2.5 cm from the centre of the scale intersection. Both were used as separate controls. All samples were cultured at 28 °C for 5–7 days, and the growth of TR4 and *Bacillus* strains were observed, the inhibition rate (%) of each strain was calculated using the formula: The inhibition rate (%) = D_dc_/D_sc_ (D_dc_: TR4 colony diameter in dual culture; D_sc_: TR4 colony diameter in single culture). Each treatment was repeated six times and samples were collected.

### 2.3. The Effects of Bacillus Strains N67, YN1282-2, WBN06, HN04 and G9R-3 on the Morphology of TR4 Mycelium

The newly-emerged TR4 hyphae near the colony edge of the biocontrol bacteria were picked out with a sterilized toothpick while being dual-cultured with TR4 for 7 days. The morphology of the TR4 mycelia was observed by light microscope at 40-times magnification (Nikon, eclipse E200), and the single-cultured newly-emerged TR4 hyphae were taken as the control, and any morphological changes of those TR4 mycelia between the TR4 single- and dual-cultures with five strains of biocontrol bacteria, respectively, were compared.

### 2.4. PCR Amplification of Biocontrol-Related Genes in Bacillus Strains

The *Bacillus* strains stored on the inclined test-tube surface were taken from the refrigerator at 4 °C and inoculated on the nutrient agar (NA, formula of NA: peptone 10 g, beef extract 3 g, sodium chloride 5 g, agar 15 g, Water 1 L, pH = 7) medium plate by the streak plate method. The strains were cultured overnight at 37 °C. The preparation of *Bacillus* genomic DNA was performed according to the instructions of Lysis Buffer for Microorganism to Direct PCR (TaKaRa, Dalian, China). The *Bacillus* monoclonal colony was selected and placed into a microtube contained with 50 μL Lysis Buffer. Then, 1–5 μL of supernatant lysis buffer were used as the template for subsequent PCR tests.

The genomic DNA of *Bacillus* strains N67, YN1282-2, WBN06, HN04 and G9R-3 was amplified, respectively, by PCR using 13 pairs of published primers as shown in Table 1. The amplification system (25 μL) comprised: 22 μL of Golden Star T6 super PCR mix (TsingKE, Beijing, China), 1 μL of upstream primer, 1 μL of downstream primer and 1 μL of template DNA. The reactions were run at 98 °C for 2 min, followed by 40 cycles of 98 °C for 10 s, 56–58 °C for 15 s and 72 °C for 30 s, with the total extending for 2 min. A no-template reaction that replaced DNA with ddH2O was used as negative control. Agarose gel electrophoresis (10 g/L) was used to detect the amplified products.

### 2.5. Comparison and Improvement of Bacillus RNA Extraction Methods and cDNA Synthesis

*B**acillus* total RNA was extracted according to the instructions of the TaKaRa MiniBEST Universal RNA Extraction Kit (TaKaRa, Dalian, China), the Eastep Super Total RNA Extraction Kit (Promega, Beijing, China) and RNAiso Plus (TaKaRa, Dalian, China). Agarose gel-electrophoresis and a nanophotometer (IMPLEN) were used to detect and compare the concentration and quality of RNA extracted by different methods.

*Bacillus* mRNA was extracted by the improved RNAiso Plus method, with slight modifications according to the RNAiso Plus instructions. First, *Bacillus* inocula were pretreated before adding RNAiso Plus as follows: after collecting the cells, the cells were washed with 2 mL 0.1% diethyl pyrocarbonate (DEPC) sterilizing water and fully dispersed with the pipette tips, then the cells were precipitated with centrifuging at 10,000 r/min at 4 °C for 1 min. The above treatment was repeated twice. Then, lysozyme (Solarbio, Beijing, China) was used to rupture the *Bacillus* cell walls as follows: 1 mL sterilized 0.1% DEPC water was added into the cells precipitation, mixed well and the lysozyme solution was prepared by adding sterilized 0.1% DEPC water to the final concentration of 4 mg/mL, incubating at 37 °C for 1.5–2 h, then centrifuging at 4 °C 10,000 r/min for 1 min, precipitating the cells and placing the precipitated cells in liquid nitrogen for 2 to 3 min. Then, 1 mL RNAiso Plus was added to the frozen precipitant and put it into a water bath at 100 °C for 15 to 25 min, finally quickly transferring it to an ice bath. To reduce the contamination of protein and phenols, the chloroform extraction process was repeated twice. The rest of the RNAiso Plus method followed the instructions’ steps. On the final step, anhydrous ethanol was used to wash the precipitation for 2 to 3 times. mRNA samples with A260/A280 and A260/A230 of 1.9 to 2.1 and 2.0 to 2.4 (measured by nanophotometer, IMPLEN), respectively, were used as templates for the RT-qPCR process.

### 2.6. RT-qPCR Detection of Expression Levels of Genes Associated with TR4 Biocontrol Activities in Bacillus

Each *Bacillus* strain of N67, YN1282-2, WBN06, HN04, G9R-3 and TR4 was placed in single or dual-culture. Samples were taken after five days, when TR4 mycelia filled the whole petri dish. Single cultures of each strain were used as controls. For sampling of *B**acillus* dual-cultured with TR4, a sterilized toothpick was used to collect cells from the 1 cm border around edge of the *Bacillus* colonies, and again the same sampling position for single cultured *Bacillus*. Sampling positions of single-cultured *Bacillus,* or for dual-cultured *Bacillus* with TR4 are shown in Figure 1b.

Primer Express (version 3.0; Applied Biosystems, Foster City, CA) was used to design the specific RT-qPCR primers (Table 2) based on 12 biocontrol-related genes in *B**acillus.* The *Rop B* (RNA polymerase β subunit) was used as a housekeeping gene for RT-qPCR detection [26] (Table 2).

A NovoScript^®^ Plus All-in-one 1st Strand cDNA Synthesis SuperMix (gDNA Purge) (Novoprotein, Shanghai, China) kit was used to reverse-transcribe the *Bacillus* total RNA into cDNA, in accordance with the kit instructions. The qPCR was performed on a CFX96^TM^ Real-Time System (Bio-Rad Laboratories, Hercules, CA, USA), and a NovoScript^®^ SYBR qPCR SuperMix Plus (Novoprotein, Shanghai, China) kit was used to detect the expression levels of genes associated with TR4 biocontrol activities in the five *Bacillus* strains namely N67, YN1282-2, WBN06, HN04 and G9R-3. The *Rop B* (RNA polymerase β subunit) was used as the housekeeper gene for all reactions. A 1-4-8-16-32-64-fold dilution gradient of each *Bacillus* strain cDNA was used to establish a relative standard curve and a dissolution curve for determining the amplification efficiency and the specificity of each pair of primers. Each pair of primers had a single peak for dissolution curve and the slope of the standard curve was greater than 0.99, and exhibited 90% to 110% amplification efficiency in all reactions. The 2^−^ΔΔCt method was used to analyse data describing genes’ relative expression levels. No less than 6 repetitions of single- or with TR4 dual-culture treatments for each *Bacillus* strain were completed, and each reaction was run in triplicate.

### 2.7. Statistical Analyses

Statistical differences of gene expression levels in each *Bacillus* strain between single or dual-cultivated with TR4 were analysed with the Student’s *t*-test (*p* < 0.05). All analyses were performed using software package SPSS 19.0 (IBM, Armonk, NY, USA).

## 3. Results

### 3.1. The Dual-Culture of Five Bacillus Strains against TR4

The dual-culture test results show that all five *Bacillus* strains have a pronounced antagonistic effect against TR4. For 7 days after dual-culture tests, compared with singly-cultured TR4, the growth of TR4 mycelia were significantly inhibited by strains N67, YN1282-2, WBN06, HN04, G9R-3 with the inhibition rate (%) of 72.12%, 79.63%, 74.71, 72.53% and 74.11%, respectively. There was a distinct inhibition zone between the *Bacillus* colony and TR4 colony shown as Figure 2a. Comparing with the diameter of the singly-cultured TR4 colony, the diameter of the TR4 colony culturing with five *Bacillus* strains was dramatically reduced (Figure 2b), which suggested that the antagonistic effects of the five *Bacillus* strains on TR4 were all obvious and could be used in the next step of experiment for its functional gene characterization.

### 3.2. Bacillus Strains Inhibit the Growth of TR4 Hyphae and Causes Hyphae Malformation

Compared with the hyphal morphology of single-cultured TR4, N67 induced obvious TR4 hyphal swelling, with shortened internodes during the dual-culture test for 7 days (Figure 3a, arrow 1). Hyphal tips were expanded and vesicular (Figure 3a, arrow 2) and the hyphae expanded in the middle (Figure 3a, arrow 3). YN1282-2 resulted in the enlargement of the TR4 hyphal tips (Figure 3b, arrow 2) and expansion in the middle of TR4 hyphae (Figure 3b, arrow 3). WBN06 resulted in the enlargement of the top of TR4 hyphae (Figure 3c, arrow 2), swelling of TR4 hyphae, shortening of hyphal internodes (Figure 3c, arrow 3) and formation of vesicles (Figure 3c, arrow 4). HN04 caused TR4 hyphal swelling (Figure 3d, arrow 1), chlamydospore formation (Figure 3d, arrow 5) and the hyphae to form vesicles (Figure 3d, arrow 4). G9R-3 caused TR4 hyphal expansion with shortened internodes (Figure 3e, arrow 1). The apex of the hyphae was enlarged (Figure 3e, arrow 2), and expanded to form vesicles (Figure 3e, arrow 4).

### 3.3. PCR Amplification Results of Biocontrol Related Genes in Biocontrol Bacillus Strains

All seven genes of *srfAA*, *fend, ituC, yngG, yndJ, yndJ* and *dhb* responsible for syntheses of non-ribosomal peptide synthetases (NRPS) metabolites were found in all five strains. The other three genes of *bae*, *dfn*, *bac* involved in synthesis of three polytide synthetases (PKS) metabolites were also found in all five strains. However, the macrolactin synthase gene *mln* was only detected in WBN06 and YN1282-2. All the five *Bacillus* strains contain the gene *bioA*, essential for Biotin synthesis (Table 3). However, the gene *sboA* which is involved in subtilosin synthesis was not detected in all five *Bacillus* strains.

### 3.4. An Improved RNAiso Plus Method Suitable for Extraction of Total RNA from Bacillus Strains

Three kits were used to extract the total RNA of the *B**acillus* strains and the concentration and purity of RNA were tested by nanophotometer. The general criteria for assessing RNA purity were the proportions of A260/A280 and A260/A230, which were within the range of 2.00 ± 0.10 and 2.00–2.40, respectively. If the proportion of A260/A280 is less than 2.00, it indicates that protein contamination existes in the RNA sample. If the proportion of A260/A230 is less than 2.00, the samples contain the remaining guanidine isothiocyanate and β-mercaptoethanol from the kit reagent. However, the results show that three kinds of RNA extraction kits could not be effective in extraction of high concentrations and qualified total RNA from *Bacillus*, as shown in Table 3. The quality of RNA extracted by the three kits were beyond the standard range required for evaluating the RNA purity, where the proportion of A260/A280 and A260/A230 of most samples were less than 2.00, indicating contaminating proteins, polysaccharides and phenols in the samples. Moreover, the concentrations of RNA extracted by the three kits were generally very low, and only the RNAiso Plus method could extract RNA with a higher concentration than the other two kits. In summary, for tested *Bacillus* strains, the extraction of total RNA, none of above three kits provided an adequate protocol (Table 4).

Some critical steps were added to the traditional RNAiso Plus kit: washing the cells with DEPC water, lysozyme was used to lysate the cells, removing the rRNA and tRNA by heating, repeated the chloroform extraction process for twice and washing the final precipitation with anhydrous ethanol for three times. By modifying the RNAiso Plus kit for total *Bacillus* RNA extraction, the proportion of A260/A280 and A260/A230 for all samples were within standard range, and RNA concentrations were significantly increased to be more than 500 ng/uL (Table 4). In addition, the 23S and 16S rRNA of the total RNA were successfully removed by heating, so the clear 23S and 16S rRNA bands could not be visible in the electrophoresis bands, and only the 5S rRNA bands were retained (Figure 4). The above improvements indicate that the high-quality *Bacillus* RNA extracted by modified RNAiso Plus method can be used in subsequent molecular experiments.

### 3.5. The Expression Levels of Biocontrol Related Genes in Bacillus Strains

Different *Bacillus* strains have various NRPS gene-expression profiles against TR4. The expression levels of *srfAA* in *Bacillus* strains of G9R-3 and N67 were significantly up-regulated after being dual-cultured with TR4 (Figure 5a,e), but no significant difference was observed in HN04, YN1282-2 and WBN06 during singly or dual-cultured with TR4 (Figure 5b–d). The expression levels of *fenD*, *ituC* and *yndJ* were significantly regulated by TR4 in all the five strains (Figure 5a–e). The expression of *bamD* in G9R-3 and HN04 could not be induced by TR4 (Figure 5d,e), but it was significantly induced and up-regulated in three other strains (Figure 5a–c). There was no significant difference in the expression levels of *yngG* in WBN06 between singly or dual-cultured with TR4 (Figure 5c), but it was significantly induced by TR4 in the four other strains (Figure 5a,b,d,e). The expressions of *dhb* were significantly up-regulated by 2.48 and 6.55-fold in N67 and YN1282-2 after dual-cultured with TR4 compared with that in single cultured cells, respectively (Figure 5c,e). However, the expressions of *dhb* in WBN06 and G9R-3 were significantly down-regulated (Figure 5a,b), and there were no significant differences for the expressions of *dhb* in HN04 between single-cultured *Bacillus* and dual-cultured with TR4 (Figure 5d). The *p*-value for Student’s *t*-test which was used to compare the difference of NRPS genes expression between each strain single- and dual-cultured with TR4 is shown in Appendix A. The results indicate that different strains of *B**acillus* may adopt different NRPS strategies against TR4.

The up-regulation patterns of all NRPS genes also varied greatly among different *Bacillus* strains when antagonizing TR4. The *yndJ* was the most significantly up-regulated NRPS gene in YN1282-2 during dual-culture with TR4, which was 6.39-fold higher than that in the singly-cultured sample (Figure 5a). The most significantly up-regulated NRPS genes in WBN06 and G9R-3 were *yngG* and *bamD*, which were up-regulated 45.28-fold and 46.03-fold, respectively (Figure 5c,e). However, the most significantly up-regulated NRPS genes in N67 and HN04 were *fenD* and *yngG* (Figure 5a,d). Our results show that the most significant up-regulated genes in each *Bacillus* strain during dual-cultured with TR4 are most likely to be NRPS key genes (Figure 5).

Different *Bacillus* strains display different PKS gene-expression profiles and patterns against TR4. In HN04 and WBN06 (Figure 6c,d), there was no significant difference in expression levels of all KPS genes among each strain during single- or dual-cultures with TR4. The expression levels of *mln*, *bae* in YN1282-2 and *bae*, *dfn* in N67 were significantly up-regulated by TR4 induction, respectively. *mln* and *dfn* were the most significantly up-regulated genes in YN1282-2 and N67, which were increased 13.56-fold and 4.34-fold after dual-culturing with TR4, respectively (Figure 6a,b). The expression level of *bae* in G9R-3 was significantly down-regulated by TR4 (Figure 6e). *p*-value for Student’s *t*-test which was used to compare the difference of PKS genes expression between each strain single and dual-cultured with TR4 was shown in Appendix A. The above results indicated again that different *Bacillus* strains may take different strategies against TR4.

During dual-culture with TR4, the expressions of *bioA* in N67, YN1282-2 and HN04 were significantly induced by TR4. However, it was significantly down-regulated in WBN06 with that in single culture. In addition, there was no significant difference in G9R-3 between single and dual-culture (Figure 7). *p*-values for Student’s *t*-test, which was used to compare the difference of *bioA* expressions between each strain single and dual-cultured with TR4, are shown in the Appendix A.

## 4. Discussion

In view of the significant potential role of the Bacillus group in controlling Fusarium wilt of banana, a deeper understanding of TR4 biocontrol mechanisms in Bacillus will contribute to the more efficient development and utilization of active Bacillus strains. In this study, all five biocontrol bacterial strains: N67, YN1282-2, WBN06, HN04 and G9R-3, could effectively cause TR4 mycelium deformity and inhibit mycelium growth (Figure 4). Therefore, these could inhibit the reproduction of pathogenic fungi as a next step. Lipopeptide antibiotics synthesized in Bacillus have played an important role in antifungal action. The substances that exert antifungal effects are mainly lipopeptides, of which fengycin and iturin have been proven to have strong antifungal activity, causing teratogenic effects on the pathogenic fungus mycelia, inhibiting spore germination, destroying membrane structure and resulting in metabolic material leakage [27,28,29,30].

Chen et al. found that surfactin can effectively enhance the inhibition ability of fengycin against fungi, and there is a synergistic effect between the two substances [31]. *B. amyloliquefaciens* SQR9 contains two lipopeptides, bacillomycin D and fengycin, which can effectively antagonize the banana Fusarium wilt fungus [32]. While polyketides such as surfactin, bacillibactin, difficidin and bacillaene display antibacterial activity, which can change the cell surface hydrophobicity and damage the integrity of the bacterial biofilm [33,34,35,36]. B. amyloliquefaciens BEB17 crude extract containing the sterile polyketides of bacillibactin, bacillaene and difficidin effectively inhibits the growth of Foc TR4 mycelia [37]. Therefore, it is speculated that the antifungal effect of the five bacillus strains evaluated in this study may be due to the presence of the key biosynthesis genes of these biocontrol substances in their genomes, resulting in the production of a variety of biocontrol substances with significant antagonistic effects against TR4. All previous studies report using only one single strain in their research. In this study, we took five strains from two species of Bacillus for systemic assessment of its genetic markers present/absent at genomic level and expression profiles at transcriptomic level during dual-culture with TR4. Strains WBN06 and YN1282-2 contain a full set of biocontrol genes except for *sboA*. Interestingly, the dual-culture test also showed that these two strains exerted a slightly better inhibition of TR4 compared to the other three strains (Figure 3). Compared with traditional confrontation and pot experiments, PCR can quickly evaluate the biocontrol potential of these bacterial strains. However, to assess if the biocontrol potential of these strains could be evaluated by the number of genetic marker genes requires further research. In addition, the synthesis of these biocontrol substances in each strain of antagonistic TR4 needs further analysis in combination with liquid chromatography–mass spectrometry (LC-MS) as a next research step.

Generating complete and high-quality RNA is the basic condition for gene-expression study by RT-qPCR [38]. The extraction of total bacterial RNA is more difficult than extracting that of eukaryotes [39]. The *Bacillus* group are Gram-positive bacteria with thick cell walls and the peptidoglycan is tightly cross-linked [40]. Most species in the *Bacillus* genus have a strong extracellular enzyme secretion system, which can secrete a variety of enzymes into the exocytosis including RNase. In addition, the combination of cellular inclusions with RNA forming the insoluble substances (polysaccharide polyphenols) makes RNA isolation much more difficult [41]. In this study, although reasonable amounts of RNA could be extracted by three conventional kits, there were still problems, such as protein contamination residues. Therefore, the team made a series of improvements to develop a better kit using the RNAiso Plus-based method. By repeatedly washing the cells three times with DEPC water, the endogenous RNase secreted by the *Bacillus* was effectively removed, and RNA degradation minimized. Lysozyme prepared with DEPC water was used to rupture the cell wall to release intact RNA and increase the concentration of extracted RNA. Usually in the cells, rRNA, tRNA and mRNA account for about 80%, 10% to 15% and 1% to 5% respectively, of the total RNA [40]. For many subsequent molecular biology studies, rRNA and tRNA are usually useless, therefore, researchers often remove rRNA and tRNA from total RNA in order to be more effective for downstream experiments [39]. Due to the broad abundance and various sizes of mRNA, no bands are usually visible during electrophoresis. However, we can judge the quality of the mRNA according to the integrity of the electrophoresis bands. In this study, the heating method was used to remove the 23S and 16S rRNA bands in total RNA, while keeping 5S rRNA in which could be detected with bands in electrophoresis. The results of electrophoresis showed that the obtained RNA bands were sharp and complete without degradation (Figure 5). Therefore, the RNA extracted using the modified protocol was able meet the requirements of RT-qPCR. The high-quality RNA template can be used not only for RT-qPCR, but also for cDNA library construction, northern hybridization, chip hybridization and other studies proposed in our next research steps.

There are few studies utilizing RT-qPCR to investigate expression of genes associated with synthesizing biological control substances [33]. However, there are no other studies on TR4 antagonistic biocontrol strains. Therefore, in this study, 12 *Bacillus* biocontrol marker genes including seven NRPS genes: *srfAA*, *fenD*, *ituC*, *bamD, yngG*, *yndJ* and *dhbA* and 5 PKS genes: *mln*, *bae*, *dfn*, *bac* and *bioA* were tested in the genome of five biocontrol bacteria strains during single or dual-cultures with TR4. RT-qPCR results showed that the expression profiles of NRPs and PKs genes vary in different *Bacillus* strains when against TR4. Several studies have shown that individual or combined lipopeptide antibiotics in *Bacillus* have contributed to biocontrol of other plant pathogens [9,13,36,42]. In addition, the types and amounts of inhibitory substances secreted by *Bacillus* strains vary in different cases [21]. Our results also further confirm that different strains of *Bacillus* may have different antagonistic strategies against TR4. Among the five strains, HN04 and G9R-3 are endophytes while the other three strains are rhizospheric bacteria derived from two main banana producing provinces of China. The gene *bamD*, as the key biosynthetic enzyme gene of bacillomycin D, was only significantly up-regulated by TR4 in three rhizobacteria, but not in the endophytes. Another study has shown that *B. amyloliquefaciens* SQR9 mutant not only reduced the inhibition efficiency of Cucumber Fusarium wilt to 40%, but also affected the formation of biofilms and the colonization of the strain [32]. However, whether bacillomycin D is the main difference between endophytic and rhizospheric bacteria in antagonizing TR4 must be further studied. The genes *yndJ* and *yngG* encode two kinds of lipopeptides with unknown functions, which widely exist in the genome of multiple biocontrol strains [21]. Our results show that the expression of these two genes in five biocontrol strains could be significantly induced by TR4 (except for *yngG* in WBN06), indicating that two lipopeptide substances may play an important role in antagonizing TR4. The gene *bioA* was not previously associated with biological control [21], and its expression was not affected by TR4 dual-culture with G9R-3. Surprisingly, its expression was significantly up- or down-regulated by TR4 with the other four strains. These results indicated that *bioA* also plays an important role in the process of biological control of TR4. There are many kinds of biocontrol mechanisms in biocontrol bacteria, among which bacillibactin is related to nutrition competition, and competes with the pathogenic fungus for iron. The expression of *dhb* which related to the synthesis of bacillibactin was not affected by TR4 in HN04, but was significantly up- or down-regulated in the other four biocontrol strains. Determining whether this is due to a synergistic or competitive relationship between nutrient competition and antagonism needs further exploration.

With the development of the biocontrol agents, biosecurity of *Bacillus* should be taken into account. The safety of biocontrol agents and its effect on ecosystem are the premise of field application. In this study, five *Bacillus* strains have the potential to synthesize NRPS, PKS and other antibacterial compounds and have good antagonistic effect on TR4, but whether these active compounds are safe to human beings and whether they will affect the ecosystem such as other crops and soil microbial communities is still needed for further assessment. Therefore, we will systematically evaluate the biosecurity of these strains to provide a theoretical basis for the development of safe and effective biocontrol agents in next step.

We have developed a reliable fluorescence transformation system of all five *Bacillus* strains for monitoring its colonization in banana for TR4 biological control [43]. In order to categorize the biological functions of these *Bacillus* stains, knocking-out these specific genes will be the next step in our research for verifying the involvement of these genes.

## 5. Conclusions

In conclusion, five *Bacillus* strains: WBN06, HN04, YN1282-2, N67, G9R-3, display strong antibiotic activity against TR4 both in vitro and in vivo. In this work, using conventional PCR methods with primer targets for 13 *Bacillus* biocontrol marker genes including 7 NRPS genes: *srfAA*, *fenD*, *ituC*, *bamD, yngG*, *yndJ*, *dhbA* and 5 PKS genes: *mln*, *bae*, *dfn*, *bac* and *bioA*, we conducted a rapid and effective assessment for biocontrol potential of *Bacillus.* Using the RT-qPCR method with specific designed primers which target the above genes, we showed the different biocontrol gene expression profiles and patterns among these strains. This may suggest different mechanisms are involved in biocontrol of TR4. The outcome of our study offers a foundation for elucidating the biocontrol mechanisms of functional genes contained in the above strains.

## Figures and Tables

**Figure 1 jof-07-00353-f001:**
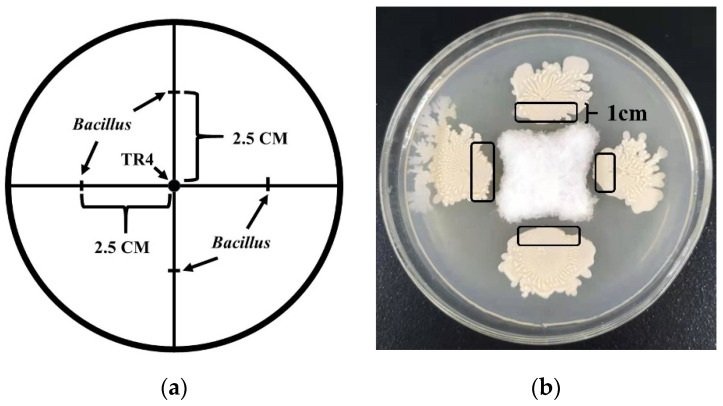
The schematic and sampling location of dual-culture test. (**a**) The sampling location for dual-culture test. The arrow points are shown to be the inoculation sites of TR4 and *Bacillus*. (**b**) The sampling location for dual-culture test. Five *Bacillus* strains and TR4 were single or dual cultivated, respectively, using dual-culture test, and samples were taken after 7 days when TR4 mycelia filled the whole petri dish. Sampling positions were circled in black ink—about 1 cm wide, and single-cultures of each strain used as controls.

**Figure 2 jof-07-00353-f002:**
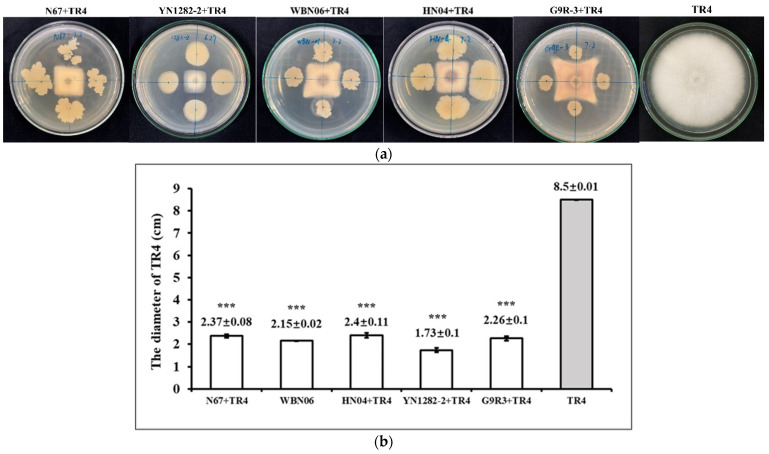
Effectiveness of five biocontrol Bacillus strains N67, WBN06, HN04, YN1282-2 and G9R-3 on TR4. (**a**) The dual-culture test of N67, YN1282-2, WBN06, HN04, G9R-3 and TR4 for 7 days; (**b**) the inhibition effect of N67, YN1282-2, WBN06, HN04, G9R-3 on TR4 at 7 days. The asterisk represents a significant difference in TR4 colonial diameter between single- and dual-cultures with biocontrol Bacillus strains (Student’s *t*-test: *p* < 0.001 “***”).

**Figure 3 jof-07-00353-f003:**
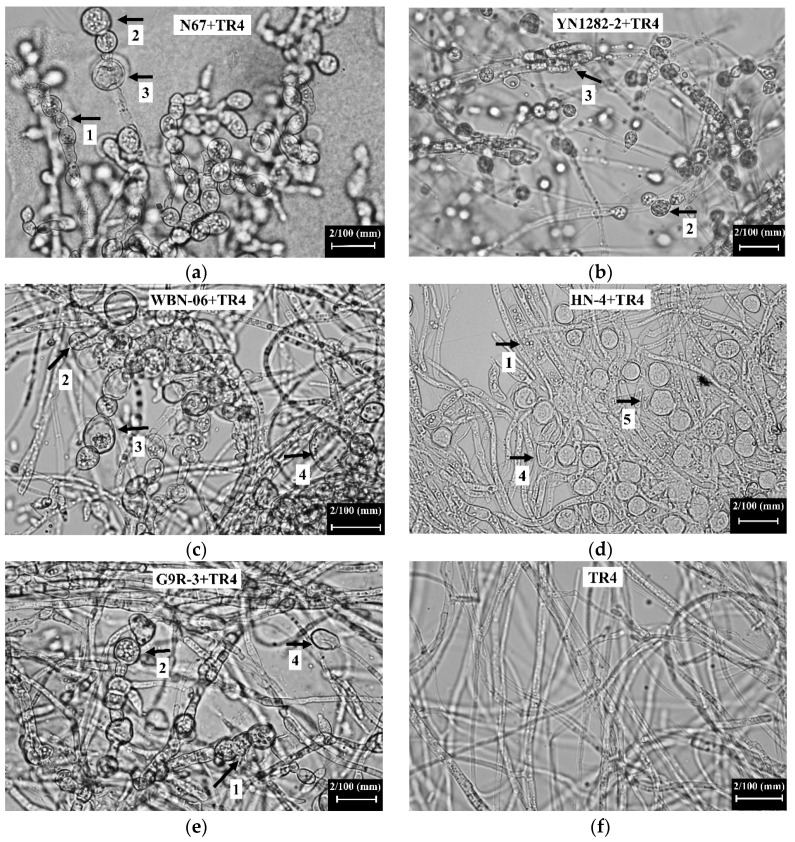
Observation of TR4 mycelial morphology in dual-culture test of N67, YN1282-2, WBN06, HN04, G9R-3 (**a**–**e**) and single-cultured TR4 (**f**) at 7 days. The white text box at the top of each picture corresponds to the mycelial morphology of TR4 when each strain was dual-cultured with TR4 for 7 days (corresponding to Figure 3(**a**–**e**)). The arrows with different labels indicate various abnormal manifestations of TR4 mycelium morphology. Arrow 1: The TR4 hyphae expanded, and the internodes shortened. Arrow 2: The top of TR4 hyphae expanded. Arrow 3: The TR4 hyphae expanded in the middle. Arrow 4: The TR4 hyphae forms vesicles. Arrow 5: The TR4 hyphae forms chlamydospores.

**Figure 4 jof-07-00353-f004:**
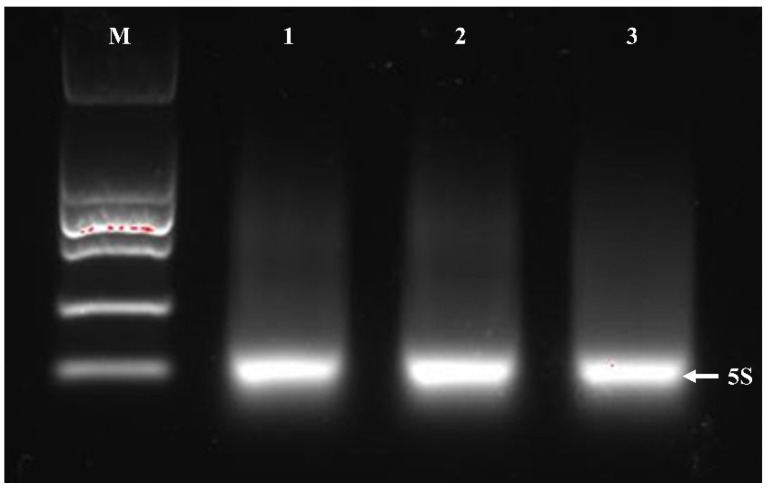
Agarose gel-electrophoresis of RNA from three *Bacillus* strains by improved RNAiso Plus method. From left to right, electrophoresis Lane M was DL15000 DNA marker, electrophoresis Lanes 1–3 were *Bacillus* sample-1 RNA, *Bacillus* sample-2 RNA and *Bacillus* sample-3 RNA. The white arrow indicates the position of the 5S electrophoresis band.

**Figure 5 jof-07-00353-f005:**
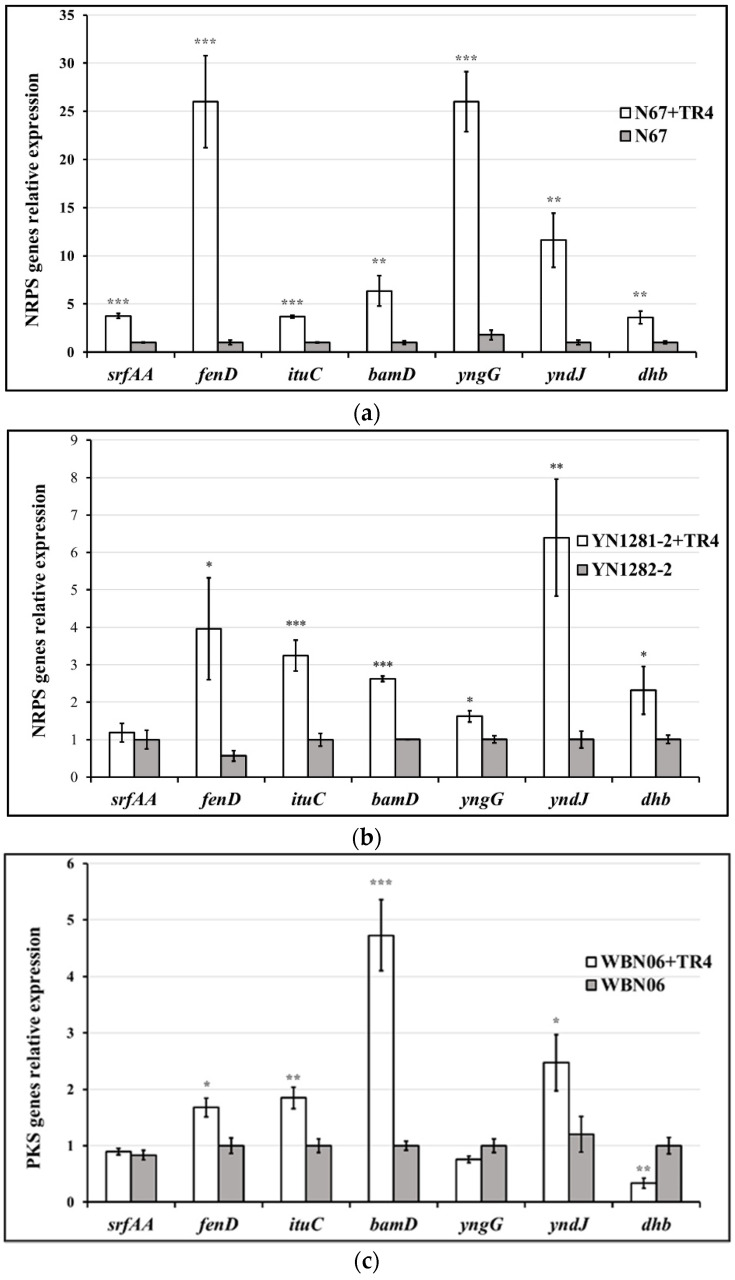
Biocontrol NRPS gene (*srfAA*, *fenD*, *ituC*, *bamD*, *yngG*, *yndJ* and *dhb*) expression profiles of five *Bacillus* strains: N67, YN1282-2, WBN06, HN04 and G9R-3. (**a**–**e**): each represent NRPS gene’s expression profiles of N67, YN1282-2, WBN06, HN04 and G9R-3, respectively. The asterisk represents a significant difference in gene expression between single and dual-cultured (Student’s *t*-test: 0.01 < *p* < 0.05 “*”, *p* < 0.01 “**”, *p* < 0.001 “***”).

**Figure 6 jof-07-00353-f006:**
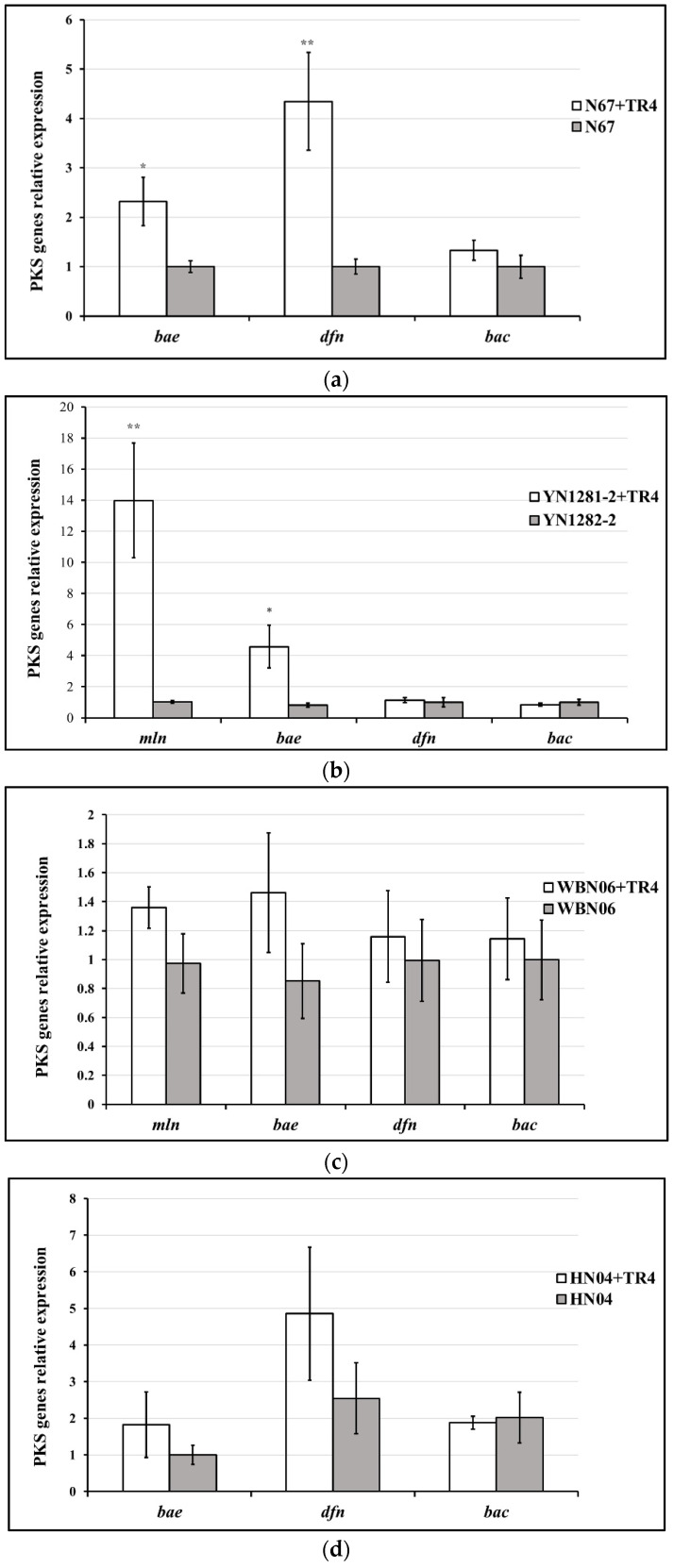
Biocontrol PKS gene (*mln*, *bae*, *dfn* and *bac*) expression profiles of five *Bacillus* strains: N67, YN1282-2, WBN06, HN04 and G9R-3. (**a**–**e**): each represent PKS gene’s expression profiles of N67, YN1282-2, WBN06, HN04 and G9R-3. The asterisk represents a significant difference in gene expression between single and dual-cultured (Student’s *t*-test: 0.01 < *p* < 0.05 “*”, *p* < 0.01 “**”, *p* < 0.001 “***”).

**Figure 7 jof-07-00353-f007:**
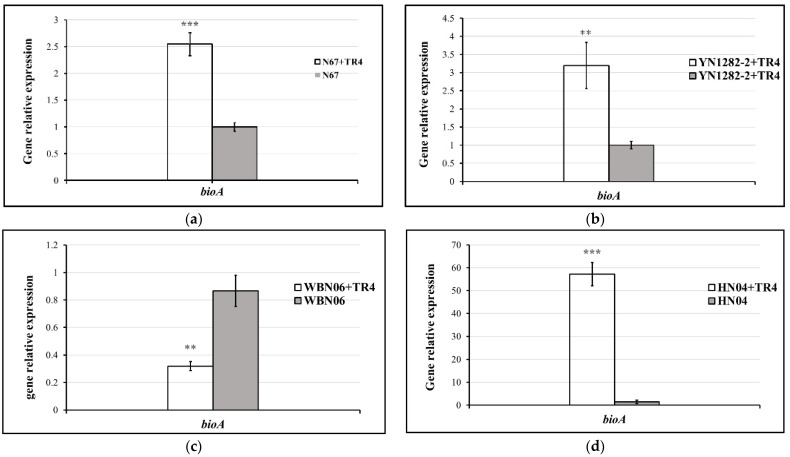
Biocontrol *bioA* gene expression profiles of five *Bacillus* strains: N67, YN1282-2, WBN06, HN04 and G9R-3. (**a**–**e**): each represent *bioA* gene’s expression levels of N67, YN1282-2, WBN06, HN04 and G9R-3. The asterisk represents a significant difference in gene expression between single and dual-cultures (Student’s *t*-test: *p* < 0.01 “**”, *p* < 0.001 “***”).

**Table 1 jof-07-00353-t001:** The specific primers from target genes derived from references.

Category	Metabolites	Synthesis Gene	Primers (5′-3′)	References
Non-ribosomal peptide synthetases (NRPS)	surfactin	*srfAA*	GAAAGAGCGGCTGCTGAAAC CCCAATATTGCCGCAATGAC	[21]
	fengycin	*fenD*	CCTGCAGAAGGAGAAGTGAAG TGCTCATCGTCTTCCGTTTC	[21]
	iturin	*ituC*	TTCACTTTTGATCTGGCGAT CGT CCG GTA CAT TTT CAC	[21]
	bacillomycine D	*bamD*	AGTCTAAGTATTGGCGAAACGA ATTATGCTGAAAGTGAAGGGCG	[24]
	yngG	*yngG*	GAACTGTCCGAAACATGTCCG CTGAGCTCTTGAACGGTCCGG	[21]
	yndJ	*yndJ*	CAGAGCGACAGCAATCACAT TGAATTTCGGTCCGCTTATC	[21]
	bacillibactin	*dhbA*	TCGGATTTCTTTTGCCACTTGG ATAACGGGCGCTCCTTCGGTTC	[24]
Polytide synthetases (PKS)	macrolactin	*mln*	GCTGATGAACTGATAACAACCG CATTACGAAGCAAAATAAAAAA	[24]
	bacillaene	*bae*	TTGTGCGGTCGTGTATGAACAG TCGCCATCGAGATTAAGAAATT	[24]
	Difficidin	*dfn*	AGTAGTTTTTCTCATCGGTCTG GGCTCCTTATATTGGGGCATTC	[24]
	Bacilysin	*bac*	TGAAGGGACAAGTAGTGAGTAC GATAGGAGACGGGTGGGATA	[24]
Ribosomal peptide synthetases (RPS)	Subtilosin	*sboA*	TCGGTTTGTAAACTTCAACTGC GTCCACTAGACAAGCGGCTC	[25]
	Biotin	*bioA*	TTCCACGGCCATTCCTATAC TTTGTCCCCTTATCCTGCAC	[21]

**Table 2 jof-07-00353-t002:** The designed specific primers and housekeeping gene for RT-qPCR.

Category	Metabolites	Synthesis Gene	Primers (5′-3′)	References
NRPS	surfactin	*srfAA*	AGCGTAAACGGCATTCAGGAG	This study
			TATGAGACGGCAGTGTTTCGG	
	fengycin	*fenD*	CATTTTAACCAGTCCGTCATGC	This study
			TCTTTTTTGCAGACAAGGCGC	
	iturin	*ituC*	AACGAATACGGGCCTACAGAG	This study
			CTTCATGCTCTTATCCAGCACG	
	bacillomycine D	*bamD*	ATTGGCGAAACGAAACATCTGC	This study
			AACATCTGATTGTGCTCACGTTC	
	yngG	*yngG*	CAGAGCGACAGCAATCACATC	This study
			ATTGCTCGGCAGGATCATACG	
	yndJ	*yndJ*	CAGAGCGACAGCAATCACATC	This study
			ATTGCTCGGCAGGATCATACG	
	bacillibactin	*dhb*	CAGTGAAATCGAGCCGATCC	This study
			TCTGAAACGGCTTTACAGCATG	
PKS	macrolactin	*mln*	CTGATGAACTGATAACAACCGAG	This study
			ACGTGCCGAAACAACGATTGG	
	bacillaene	*bae*	TGT GCG GTC GTG TAT GAA CAG	This study
			AAC GGT CTG TAT AAA TGC CGA TG	
	difficidin	*dfn*	TAT CTC AAT CGG ATC GCC GAG	This study
			ATA CGG TGC CTA ATC CGG AAG	
	bacilysin	*bac*	TGAAGGGACAAGTAGTGAGTAC	This study
			AGGCACAATTGTGTATTCCAGC	
RPS	biotin	*bioA*	GTCGCCGAAAAATCAAAAACGG	This study
			ACAAGCTCTATGCCGCACATG	
Housekeeping gene	RNA polymerase β subunit	*RopB*	AGTATCCCGTTGAAGAGTCAAAAGA	[26]
			CAAGCTGAGATACGATAACACGTTC	

**Table 3 jof-07-00353-t003:** The overview of biocontrol marker gene amplification status in tested strains of *Bacillus.*

Category	Metabolites	Synthesis Gene	N67	WBN06	HN04	YN1282-2	G9R-3
NRPS	surfactin	*srfAA*	+	+	+	+	+
	fengycin	*fenD*	+	+	+	+	+
	iturin	*ituC*	+	+	+	+	+
	bacillomycine D	*bamD*	+	+	+	+	+
	yngG	*yngG*	+	+	+	+	+
	yngJ	*yndJ*	+	+	+	+	+
	bacillibactin	*dhb*	+	+	+	+	+
PKS	macrolactin	*mln*	−	+	−	+	−
	bacillaene	*bae*	+	+	+	+	+
	difficidin	*dfn*	+	+	+	+	+
	bacilysin	*bac*	+	+	+	+	+
RPS	subtilosin	*sboA*	−	−	−	−	−
	biotin	*bioA*	+	+	+	+	+

The genomic DNA of *Bacillus* strains N67, YN1282-2, WBN06, HN04 and G9R-3 was amplified respectively by PCR using 13 pairs of published primers which target thirteen biocontrol marker genes of the *Bacillus* group. “+” represents the positive result of PCR, “−“represents the negative result of PCR.

**Table 4 jof-07-00353-t004:** Comparison of the quality and yield of total *Bacillus* RNA extracted by the different kits and the modified protocol.

Method	RNA Quality Determination	*Bacillus* Sample-1	*Bacillus* Sample-2	*Bacillus* Sample-3
Mini BEST	A260/A280	1.87	1.64	1.77
	A260/A230	1.21	1.32	1.42
	OD/(ng/μL)	16.22	9.76	27.41
Eastep Super	A260/A280	2.13	2.44	1.68
	A260/A230	1.92	2.58	1.74
	OD/(ng/μL)	24.8	34.1	33.6
RNAiso Plus	A260/A280	2.43	2.17	1.86
	A260/A230	1.94	2.28	1.71
	OD/(ng/μL)	97.71	108.92	84.3
Modified RNAiso Plus	A260/A280	2.15	2.09	2.16
	A260/A230	2.19	2.23	2.17
	OD/(ng/μL)	534.17	648.45	549.56

## Data Availability

The strains YN1282-2, N67, WBN06, HN04 and G9R-3 were registered in NCBI, GenBank Accession No. YN1282-2, N67, WBN06 is MW663765, MW672323, MW672324, MW674626 and MW674627, respectively. The data presented in this study are available in this manuscript and materials can be requested from the corresponding author.

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
