# Peer review of "A Real-Time Fluorescent Reverse Transcription Quantitative PCR Assay for Rapid Detection of Genetic Markers’ Expression Associated with Fusarium Wilt of Banana Biocontrol Activities in Bacillus"

_jof, 2021, doi:10.3390/jof7050353_

Round 1

Reviewer 1 Report

I consider that the results shown in this manuscript are important.

Author Response

Response to Reviewer 1 Comments

We were pleased to read the insightful evaluations of the reviewer 1.

Point 1: I consider that the results shown in this manuscript are important.

Response 1: Thank the reviewer 1 for the positive recommendation.

Sincerely and with our best regards,

Si-Jun Zheng on behalf of the co-authors

Reviewer 2 Report

Methodically very interesting article. It is a pity that the final conclusion does not give any fundamental answer, and the authors did not attempt to further research to unravel the topic. Nevertheless, the research are described very substantively and the results are presented clearly. From the linguistic point of view, the article is written very well.

  1. In the abstract, the authors could focus a little more attention on the description of the qPCR results to comply with the title. In the current version, more attention is paid to gene detection in traditional PCR.
  2. I propose to allocate a bit of description to the broader characteristics of TR4. Maybe virulence factors or at least identification markers in the form of ITS sequences with accession numbers are described in literature. This aspect is somewhat lacking in the article.
  3. Fig. 4 I would like to remove completely. Description in the text is sufficient.

Author Response

Response to Reviewer 2 Comments

Methodically very interesting article. It is a pity that the final conclusion does not give any fundamental answer, and the authors did not attempt to further research to unravel the topic. Nevertheless, the research are described very substantively and the results are presented clearly. From the linguistic point of view, the article is written very well.

Response: We were pleased to read the insightful evaluations of the reviewer 2. We have evaluated all comments and have prepared a careful revision of the manuscript that addresses all comments by the reviewer 2. We thank the reviewer 2 for his/her constructive comments and positive feedback.

In the abstract, the authors could focus a little more attention on the description of the qPCR results to comply with the title. In the current version, more attention is paid to gene detection in traditional PCR.

Response: Thank reviewer 2 for this kind of suggestion. We have added the text “with novel designed specific primers” linking the qPCR in the abstract.

I propose to allocate a bit of description to the broader characteristics of TR4. Maybe virulence factors or at least identification markers in the form of ITS sequences with accession numbers are described in literature. This aspect is somewhat lacking in the article.

Response: Thank reviewer 2 for this kind of suggestion. However, vegetative compatibility group is unique and better for characteristics for describing isolates of Fusarium wilt for its virulence to host genotypes. Therefore, we have added “Due to its virulence factor TR4 isolates are designated as vegetative compatibility group VCG 01213 (or VCG 01216), which originates from Indonesia and affects many banana varieties” in the introduction part in lines 44-46.

Fig. 4 I would like to remove completely. Description in the text is sufficient.

Response: Thank reviewer 2 for this suggestion. However, due to extremely difficult for extraction of total RNA from Bacillus strains following routine protocol, we have developed an improved RNAiso plus method suitable for qPCR detection for Bacillus strains. Therefore, we would like to keep Fig. 4 as for a good demonstration of our progress to comply with the title.

Sincerely and with our best regards,

Si-Jun Zheng on behalf of the co-authors

Reviewer 3 Report

Dear authors

I would ask you to include the following in your discussion:

You write that the tested Bacillus strains inhibit the spore development of the pathogenic fungus Fusarium oxysporum f. sp. cubense (especially Tropical Race 4 - TR4). Has this been found in tests on pure cultures in Petri dishes? Have pure cultures of the fungus sprayed with bacteria become sporulation poor?

Biosecurity of bananas in the future will include spraying plants and fruit with biosprays, correct? Are the bacteria tested safe for humans? Were they found to be neutral, positive or negative? Are they present in the human digestive tract (gut)?

Are those that are endophytes enough to apply once and they will be present in plants and protect them from pathogens? And others need to be applied multiple times to get full protection?

The same is true for the soil: If applied in the natural environment, are they safe for the ecosystem, especially for the biodiversity of native microorganisms in the rhizosphere?

In the introduction, the last lines (85-97) are actually a summary of the essay, meanwhile they are supposed to be space for hypotheses and justifications for conducting the research. Also, pay attention to the spelling of units, which should be separated by a space, e.g., L106 - 90 cm, L109 - 0.7 cm, L140- 10 g, L141 3 g, L154 10 g/L, L168- 10,000 r/min, L170 0.1% DEPC, or L172 - 15-2 h. There are many more such places. Also the temperature notation should be unified: now they are sometimes with spaces, e.g. L151 and L152 - 98 °C, sometimes without spaces 56-58 °C.

https://doi.org/10.3389/fmicb.2017.01438

https://doi.org/10.3389/fmicb.2018.03236

Author Response

Response to Reviewer 3 Comments

We were pleased to read the insightful evaluations of the reviewer 3. We have evaluated all comments and have prepared a careful revision of the manuscript that addresses all comments by the reviewer 3. We thank the reviewer 3 for his/her constructive comments.

Point 1: You write that the tested Bacillus strains inhibit the spore development of the pathogenic fungus Fusarium oxysporum f. sp. cubense (especially Tropical Race 4 - TR4). Has this been found in tests on pure cultures in Petri dishes? Have pure cultures of the fungus sprayed with bacteria become sporulation poor? 

Response 1: Thanks reviewer for this kind of question. In this study, we mainly focused on the inhibition of TR4 mycelium growth and development by testing Bacillus strains in dual-culture test (Figure 2-3), but the related effects of tested Bacillus strains on TR4 spore development were not involved. Other researchers have proved that some Bacillus strains can inhibit the spore germination of pathogens, mainly involving fengycin and iturin produced by Bacillus (in lines 397-400). In this study, tested Bacillus strains had the potential to synthesize fengycin and iturin, and could strongly inhibit the mycelial growth of TR4, it is speculated that these strains also have the potential to inhibit the spore development and germination of TR4. However, we did not carry out relevant verification. We will carry out this part of experiments systematically in the future.

Point 2: Biosecurity of bananas in the future will include spraying plants and fruit with biosprays, correct? Are the bacteria tested safe for humans? Were they found to be neutral, positive or negative? Are they present in the human digestive tract (gut)?

Response 2: Thanks reviewer for raising these good comments, which help us considering this important issue. As for the biosecurity of Bacillus, we added it in the discussion section. In practice bioproducts are mainly applied to young banana plantlets in early stage in nursing and also in field after transplanting into field. After fruits setting it is usually not necessary to apply for it anymore. The biosecurity of biocontrol agents and their effects on banana and other crop or microorganisms in the field are the premise to determine whether they can be released in the field.

In theory, these bacteria are safe to human beings because all isolates from this study indicated in lines 101-103 exist naturally without any genetic modification. However, the safety of human beings in actual production and application needs to be further evaluated by animal experiments and clinical trials.

In current study, we did not test those bacteria for humans. Also, we have no knowledge whether they exist in the human digestive system. Therefore, it will include those tests in our future study.

Point 3: Are those that are endophytes enough to apply once and they will be present in plants and protect them from pathogens? And others need to be applied multiple times to get full protection?

Response 2: Thanks reviewer for this kind of question. We usually use a certain concentration of biocontrol bacteria solution to drench the plant in pot, so that bacteria can colonize in the banana rhizosphere or plants. In pot experiment, drenching with biocontrol bacteria solution once can protect plants and reduce the incidence of TR4. However, it is needed for further study in the field.

Point 4: The same is true for the soil: If applied in the natural environment, are they safe for the ecosystem, especially for the biodiversity of native microorganisms in the rhizosphere?

Response 4: Thanks reviewer again for your concern. Rhizosphere and endophyte bacteria should be safe for the soil ecosystem because they are isolated from banana plants and rhizospheric soil of banana plantations and will not affect the diversity of soil microorganisms.

Point 5: In the introduction, the last lines (85-97) are actually a summary of the essay, meanwhile they are supposed to be space for hypotheses and justifications for conducting the research. Also, pay attention to the spelling of units, which should be separated by a space, e.g., L106 - 90 cm, L109 - 0.7 cm, L140- 10 g, L141 3 g, L154 10 g/L, L168- 10,000 r/min, L170 0.1% DEPC, or L172 - 15-2 h. There are many more such places. Also the temperature notation should be unified: now they are sometimes with spaces, e.g. L151 and L152 - 98 °C, sometimes without spaces 56-58 °C.

Response 4: Thanks reviewer for good suggestion, we have rephrased lines (85-97) into hypotheses formulation and necessities for conducting this research.

The spelling of units has been thoroughly formatted in above indicated lines accordingly.

Sincerely and with our best regards,

Si-Jun Zheng on behalf of the co-authors
